# WSN Deployment Strategy for Real 3D Terrain Coverage Based on Greedy Algorithm with DEM Probability Coverage Model

**Wendi Fu, Yan Yang *, Guoqi Hong and Jing Hou**

School of Electronic Information, Northwestern Polytechnical University, Xi'an 710000, China;
2019261787@mail.nwpu.edu.cn (W.F.); honggq@mail.nwpu.edu.cn (G.H.); jhou0825@nwpu.edu.cn (J.H.)
* Correspondence: yangyan7003@nwpu.edu.cn; Tel.: +86-133-7900-0828

**Abstract:** The key to the study of node deployment in Wireless Sensor Networks (WSN) is to find the appropriate location of the WSN nodes and reduce the cost of network deployment while meeting the monitoring requirements in the covered area. This paper proposes a WSN node deployment algorithm based on real 3D terrain, which provides an effective solution to the surface-covering problem. First of all, actual geographic elevation data is adopted to conduct surface modeling. The model can vividly reflect the real terrain characteristics of the area to be deployed and make the deployment plan more visible and easy to adjust. Secondly, a probabilistic coverage model based on DEM (Digital Elevation Model) data is proposed. Based on the traditional spherical coverage model, the influence of signal attenuation and terrain occlusion on the coverage model is added to make the deployment model closer to reality. Finally, the Greedy algorithm based on grid scanning is used to deploy nodes. Simulation results show that the proposed algorithm can effectively improve the coverage rate, reduce the deployment cost, and reduce the time and space complexity in solving the WSN node deployment problem under the complex 3D land surface model, which verifies the effectiveness of the proposed algorithm.

**Keywords:** wireless sensor networks; 3D surface covering; node deployment; real 3D terrain modeling; covering model; Greedy algorithm

## 1. Introduction

Wireless Sensor Networks (WSNs) are composed of wireless sensor network nodes, which transmit the information collected in the detection area to the sink node in a self-organizing and multi-hop way, thereby actualizing the monitoring of the information in the detection area. With the development of the wireless network, WSN has been widely used in fine agriculture and smart factory and disaster prevention and reduction [1]. In many types of research on wireless sensor networks, node deployment is one of the most fundamental problems because it is directly related to the network quality of service.

The earliest studies on the coverage of wireless sensor nodes mainly focused on the ideal two-dimensional plane [2]. The nodes that adopted the two-dimensional disk-sensing model, the ant colony algorithm (ACO) [3], and the cuckoo search algorithm (CS) [4,5] could improve the coverage of the target area in the two-dimensional plane. A 3D spherical sensing model based on 3D space coverage was proposed to study the coverage problem further [6]. Mnasri [7] compared 2D deployment with 3D deployment and proved that the latter was more complex because it could solve more constraints brought by practical problems. Boufares [8] proposed a distributed algorithm based on virtual force to actualize autonomous coverage of the sensor nodes in the 3D region. Du [9] combined distributed particle swarm optimization (DPSO) algorithm with the virtual force algorithm (VF) and demonstrated the superiority of this algorithm in solving the performance of deployment nodes in the 3D environment. Furthermore, in order to better account for the signal attenuation problem in practice, the probabilistic perception model based on disk and

sphere has been gradually used to solve the deployment problem of WSN nodes [10]. Hossain [11] discussed the influence of different sensor coverage models on the network coverage range. Hao [12] proposed a 3D coverage deployment method based on WSN probabilistic model and proved the effectiveness of the method.

The three-dimensional surface-coverage problem is a particular type of problem in WSN node deployment [13]. The WSN nodes are distributed on a three-dimensional surface to complete the task of covering the surface. The WSN coverage problem in 3D terrain is close to actuality and has a certain degree of complexity, so there are few related studies [14–17]. The general solution to the covering problem of the complex surface is as follows:

1. The functional surface is used to simulate the surface environment of the area to be deployed.
2. The three-dimensional spherical covering model is adopted as the covering model of WSN nodes.
3. Use heuristic optimization algorithms (such as PSO, GA, ACO, etc.) to complete nodes' deployment.

The current states and progress of the 3D surface-covering problem are outlined: Boufares [18] proposes a distributed deployment algorithm (3D-IDVFA-TC) based on the improved virtual force strategy, which can effectively improve the coverage rate of different complex Z-degree surface function models while ensuring network connectivity and reducing node loss. Saha and Saikia [19] proposed a method based on Delaunay triangulation and projection to optimize sensor node deployment in hilly areas. Felamban [20] obtained the deployment location of nodes through the three-dimensional filling of the truncated octahedron.

Although the above algorithms can achieve a better deployment effect in the WSN surface-coverage problem, four significant issues have yet to be addressed: first of all, the model of the area to be deployed still uses the simple surface generated by the function, which is too regular and smooth to simulate the surface characteristics of the real terrain environment. Secondly, the above solution needs to estimate the required number of nodes based on the surface function. However, due to the irregularity of the real terrain, it cannot obtain the surface function of the area and consequently cannot calculate the number of nodes. Thirdly, the coverage of nodes is primarily irregular shapes in the complex surface environment, but the traditional spherical coverage model does not consider the impact of the real terrain on coverage. Therefore, it is urgent to propose a node coverage model that fits the real surface environment better. Finally, the previous deployment algorithms are easy to fall into local optimum and slow convergence because of the large amount of surface information data in the monitoring area, and the new coverage model is more complex. Therefore, to solve the WSN deployment problem of real three-dimensional terrain more effectively, a deployment algorithm that can fully integrate terrain data and node coverage model characteristics is required.

Based on the above ideas, this paper proposes a WSN node deployment algorithm based on real three-dimensional terrain, which uses the real terrain elevation data downloaded from a network geographic database to reconstruct and model the surface. A probabilistic coverage model based on Digital Elevation Model (DEM) data is proposed, which can more truly reflect the influence of the geographical environment on the coverage of nodes. Finally, considering the efficiency of the Greedy algorithm in solving the optimal solution problem, we combine the Greedy algorithm for node deployment. Simulation results verify that the proposed coverage model can reflect the real coverage more realistically under diverse surface models with different complexity. Meanwhile, combined with the idea of the Greedy algorithm, it can quickly complete the deployment of nodes and achieve a higher coverage rate.

The contributions of this study are as follows:

- Terrain-modeling technology based on DEM data is applied to solve the WSN node deployment problem, which actualizes the visualization of the WSN node's three-dimensional surface deployment plan.
- A WSN probabilistic coverage model based on DEM data is proposed in this study. This model considers factors such as signal attenuation and terrain occlusion on the original basis, making the coverage model closer to actuality and more practical.
- The Greedy algorithm based on grid scanning is employed to solve the WSN node coverage problem, which significantly reduced the time and space complexity of the algorithm.

The remainder is assigned as follows. Section 2 introduces the methods and steps of establishing a three-dimensional real surface model. In Section 3, the proposed probabilistic coverage model of WSN based on DEM data is described in detail. In Section 4, an efficient sensor deployment algorithm on 3D real terrain is presented. Section 5 conducted a series of simulation comparison tests. Finally, a conclusion is provided in Section 6.

## 2. Real Three-Dimensional Surface Model

In this section, we will introduce the methods of constructing the real three-dimensional surface models, including the acquisition and processing of DEM data and commonly used modeling methods and comparisons. Constructing a real three-dimensional surface model can reflect the real terrain characteristics of the area to be deployed and provide a reference for subsequent node deployment decisions.

The geographic data used in this study comes from the Geospatial Data Cloud [21]. It provides a large amount of real surface data information for downloading, and the data types are rich and diverse. Among them, the Digital Elevation Model, or DEM for short, actualizes the digital simulation of the ground terrain through limited terrain elevation data (that is, the digital expression of the terrain surface morphology). It is a solid-ground model that uses a set of ordered numerical arrays to represent ground elevation. The DEM file in the image format was opened by Global Mapper, then the image was intercepted and ultimately converted into evaluation value data output.

Commonly used DEM data representation models are the contour model, regular grid model (GRID), and irregular triangular network model (TIN), as shown in Figure 1. The contour model emphasizes the topography features through contour lines, shows the steepness of the slope through the density between the lines, and can directly display the height of the terrain with elevation data. However, it cannot visually display the overall topography and geomorphology. Although the GRID model and the TIN model cannot display topographical changes in the form of data, they can intuitively reflect the general appearance of the terrain and have a higher degree of visualization.

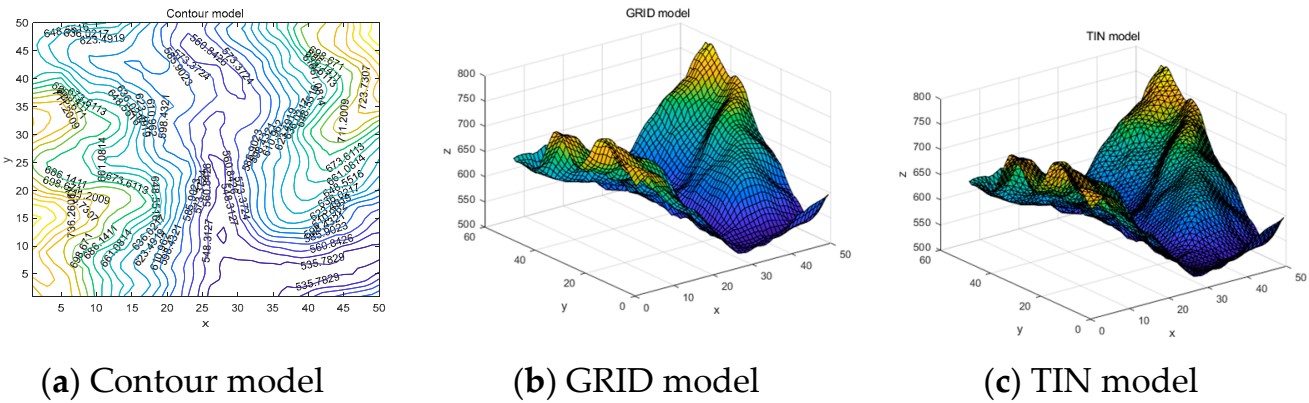

(**a**) Contour model     (**b**) GRID model     (**c**) TIN model

**Figure 1.** Contour model, GRID model, TIN model.

　　　　Both the GRID model and the TIN model have the advantages of fully expressing the structural features of the terrain. However, compared with the TIN model, the GRID model has a more straightforward structure, a smaller amount of stored data, and is more convenient for analyzing and calculating the terrain later. Therefore, we choose the GRID model to establish the surface model.

　　　　The general modeling steps of the GRID model are: first, mesh the plane area according to the number of samples of the DEM points in the horizontal and vertical directions, and then, assign the corresponding elevation values to the grid points.

　　　　The amount of DEM data used to construct the surface model is closely related to the modeling accuracy. As shown in Figure 2, the higher the DEM data volume, the higher the sampling accuracy. It means the larger the data storage and the more complicated the calculation. Selecting appropriate modeling accuracy can reduce the number of model calculations and save time and cost while meeting the coverage requirements.

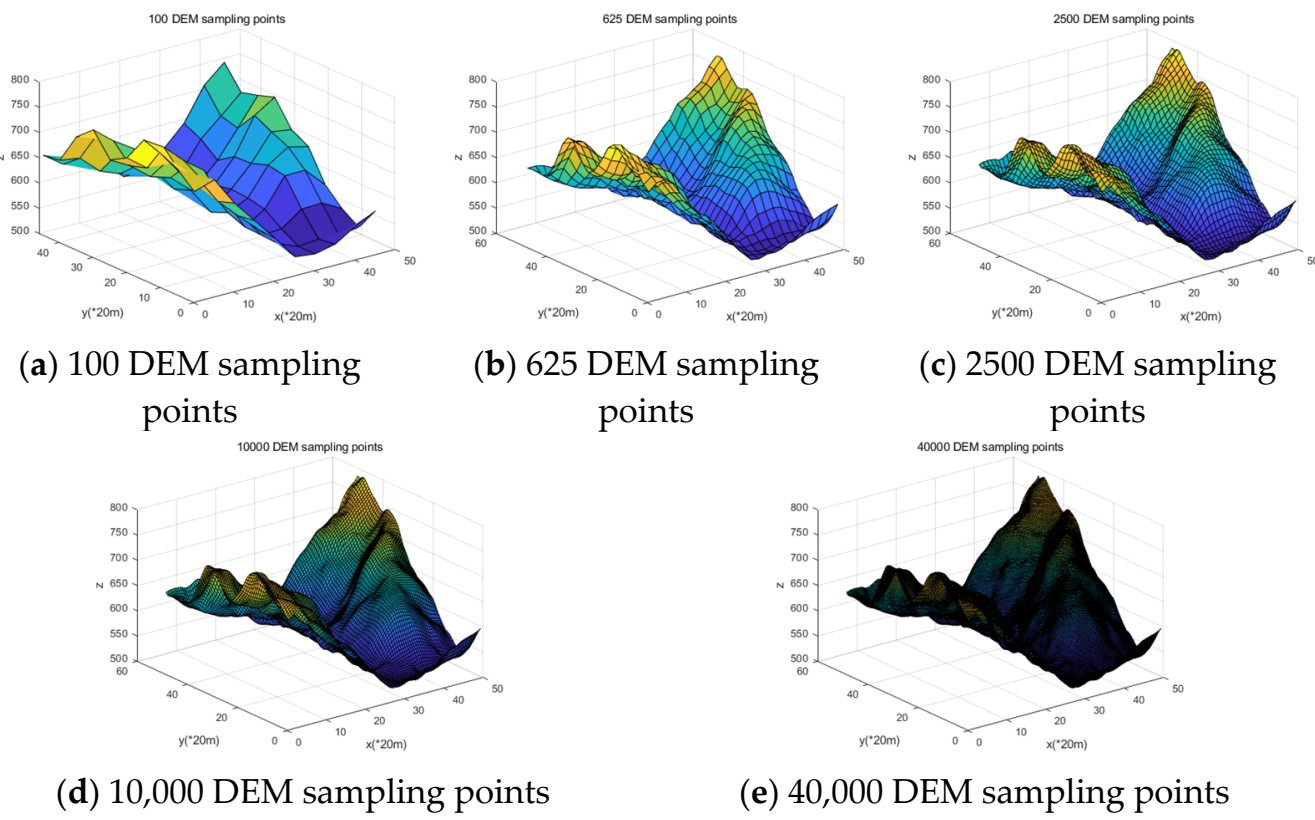

(**a**) 100 DEM sampling points　　(**b**) 625 DEM sampling points　　(**c**) 2500 DEM sampling points

(**d**) 10,000 DEM sampling points　　　　(**e**) 40,000 DEM sampling points

**Figure 2.** Comparison of DEM data volume and modeling accuracy.

## 3. Probabilistic Coverage Model of WSN Based on DEM Data

### 3.1. Problem Formulation

　　　　For the problem of WSN on surface coverage, the most commonly used node coverage model is spherical, and the coverage criterion is a single distance criterion that can only be declared to be covered if the spatial distance is smaller than the node coverage radius. In the following Formula (1):

$$P = \begin{cases} 1 & d \leq r \\ 0 & d > r \end{cases} \tag{1}$$

where $P$ is the coverage probability, $d$ is the distance from the node, and $r$ is the node coverage radius.

　　　　In practical applications, the deployment environment of WSN nodes is primarily mountainous terrain with large undulations and irregularities, where the coverage of nodes

is easily affected by the terrain. In this case, using the traditional spherical coverage model will cause an error between the real coverage and the calculated results. Suppose that the node is deployed at position 1 using the traditional spherical coverage model. In this case, point 2 can be covered in Figure 3, but the actual situation is that the terrain occludes the coverage blind area, and electromagnetic waves cannot cover the two points.

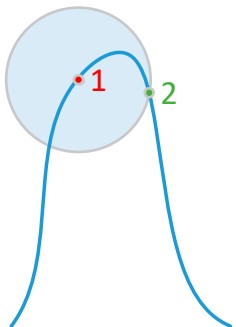

**Figure 3.** Schematic representation of the effect of terrain shading on model coverage.

In order to solve the above problem, a WSN probability coverage model based on DEM data is proposed. Based on the traditional coverage model, it considers the influence of signal attenuation and terrain occlusion factors on the coverage model, which agrees well with the perceived characteristics of nodes in practical applications.

*3.2. Coverage Model Construction*

3.2.1. The Influence of Terrain-Shading Factors on the Coverage Model

Since the continuous terrain surface is discretized into DEM data points $(x, y, z)$, then the continuous surface-coverage problem is transformed into a discrete grid point coverage problem. Now we will proceed with the following steps to solve this problem.

First, store the coordinates $(x_i, y_j, z(x_i, y_j))(i = 1, 2, \ldots, m; j = 1, 2, \ldots, n)$ of all discrete grid points on the three-dimensional surface model into the matrixes $X$, $Y$, $Z$, respectively shown in Equation (2), where the coordinate $z(x_i, y_j)$ is the elevation coordinate value corresponding to the grid point $(x_i, y_j)$:

$$
\begin{aligned}
X &= [x_1, x_2, \ldots, x_m] \\
Y &= [y_1, y_2, \ldots, y_n] \\
Z &= \begin{bmatrix}
z(x_1, y_1) & z(x_1, y_2) & \ldots & z(x_1, y_n) \\
z(x_2, y_1) & \ldots & \ldots & z(x_2, y_n) \\
\ldots & \ldots & \ldots & \ldots \\
z(x_m, y_1) & z(x_m, y_2) & \ldots & z(x_m, y_n)
\end{bmatrix}
\end{aligned} \tag{2}
$$

For determining the influence of the terrain occlusion in any sensing direction of the sensor node, it is necessary to estimate the ground surface height in this direction since the terrain surface is discretized into elevation points, as shown in Figure 4. For any sensor node with a sensing radius of $r$ on a discrete grid point, taking the node coordinates as the origin, the sensing direction angle $\theta(\theta \in [0, 2\pi])$ as the step direction, and $g/\cos\theta$ as the step length, where $g$ is the distance between discrete grid points. The elevation value of each stepped point is calculated by interpolation of the four surrounding grid points. Then, the elevation values $\{z_{\theta 1}, z_{\theta 2} \ldots z_{\theta i} \ldots z_{\theta k}\}$ for all step points within the perceptual range $r$ in the perceptual direction $\theta$ can be obtained.

Based on the approximate elevation values $\{z_{\theta 1}, z_{\theta 2} \ldots z_{\theta i} \ldots z_{\theta k}\}$, Equation (3) determines how the grid points in the sensing direction are affected by the terrain shading. $P_{ter} = 1$ means that sensor node can cover the grid points without being affected by the terrain, and $P_{ter} = 0$ implies that the grid points cannot be covered by the influence of terrain occlusion. $z_i(0 \leq i \leq n)$ is the elevation value of the corresponding step point of the

sensor node in the sensing direction, $z_0$ is the elevation value of the grid point where the sensor node is located, and $z_n$ is the farthest grid point that the sensor node can perceive in the sensing direction:

$$P_{ter} = \begin{cases} 1 & \forall z_j \leq z_i \\ 0 & \exists z_j > z_i \end{cases} (0 \leq i \leq n, 0 \leq j \leq i) \tag{3}$$

An illustrative example is shown in Figure 5; it can be calculated that the grid points $\{z_1, z_2 \ldots z_5, z_6\}$ in the figure are affected by the terrain shading. According to Equation (3), grid points 2, 4, and 6 are affected by the grid points located in front of them and have larger elevation values, which block the sensory signal of the node, resulting in coverage blind spots, as shown in Table 1.

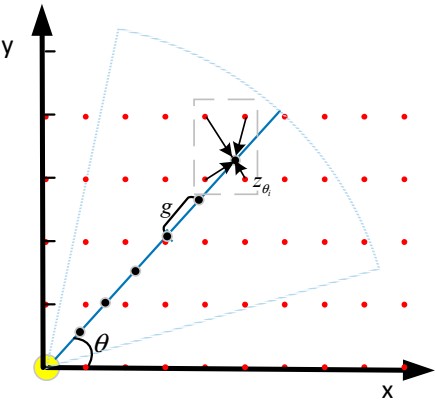

**Figure 4.** Estimation of the elevation value of the node in any sensing direction.

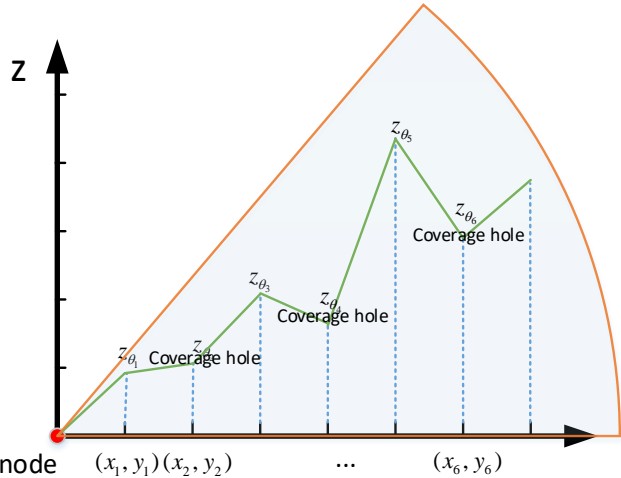

**Figure 5.** Elevation value point coverage criterion in the direction of node perception.

**Table 1.** The impact of terrain shading on corresponding grid points.

| The Number of Grid Points | Impact of Terrain Shading $P_{teri}$ |
|:---:|:---:|
| grid 1 | 1 |
| grid 2 | 0 |
| grid 3 | 1 |
| grid 4 | 0 |
| grid 5 | 1 |
| grid 6 | 0 |

By increasing the sensing direction angle $\theta$ in the range $[0, 2\pi]$, it can be determined whether each grid point in the sensing range of the node is affected by the terrain occlusion.

### 3.2.2. The Influence of Signal-Attenuation Factors on Coverage Model

Furthermore, to account for the influence of signal propagation attenuation on the coverage model, we combine the terrain shading influence criterion with the spherical probability perception model [7].

The coverage probability of the spherical probability perception model changes with the distance of the node. Mathematically, the coverage probability is shown in Equation (4). This model avoids the traditional Boolean perception model's cliff-like perception boundary and is closer to the actual propagation of the perception signal:

$$P_{sph} = \begin{cases} 1 & d \leq R - r \\ e^{-\alpha\lambda^\beta} & R - r < d < R + r \\ 0 & d \geq R + r \end{cases} \tag{4}$$

where $P_{sph}$ is the probability that can perceive a specific grid point under the spherical probability coverage model, $d$ denotes the distance between the grid point and the node, $R$ represents the maximum sensing radius when there is no signal attenuation, and $r$ is used to describe the uncertainty perception ability of the node, $\lambda = d - (R - r)$.

$\alpha, \beta$ are attenuation factors that determine the degree of attenuation of the perceived probability of a node in an uncertain sensing region with distance. The relationship between $d$ and $P_{sph}$ is shown in Figure 6. In the figure, $R$ is set to 5 and $r$ is 3. When $\alpha = \beta = 1$, the perceived probability decays faster. When $\alpha = \beta = 0.5$, the probability decays slowly. The values of $\alpha, \beta$ can be selected according to the actual situation of the node to meet different coverage requirements.

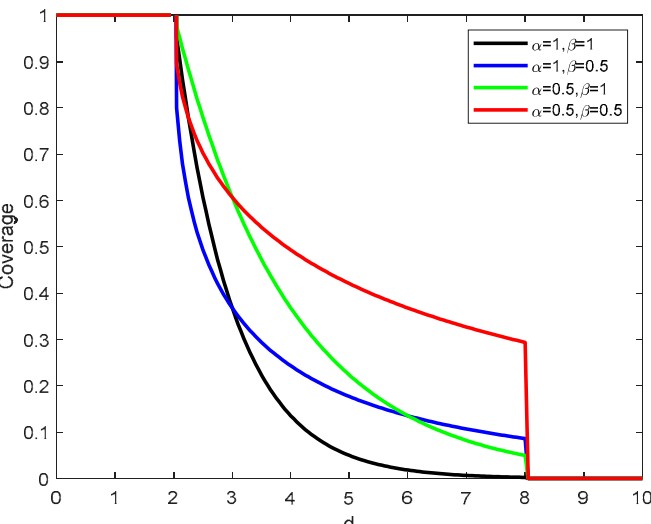

**Figure 6.** The relationship curve between coverage rate and distance under different attenuation factors.

### 3.2.3. The Influence of Signal Attenuation Factors on the Coverage Model

Finally, combining the above two models, the WSN probability attenuation coverage model based on DEM data is obtained as:

$$P = P_{ter} \cdot P_{sph} \tag{5}$$

This model comprehensively considers the influence of the terrain shading and signal attenuation on the node coverage model, so the model can more genuinely reflect the perceived effect of the node in the actual environment.

## 4. WSN Deployment Based on Grid Scanning Using the Greedy Algorithm

The Greedy algorithm is a typical heuristic algorithm. It makes the current best choice at every step, which means that it is not considering optimization as a whole, but a local optimization. However, this kind of optimal local solution is often relatively close to the optimal global solution. Greedy algorithms tend to save more time and cost than global optimization algorithms to solve problems.

A solution for the deployment of WSN nodes on real 3D terrain is proposed. The probabilistic attenuation coverage model of DEM data is adopted for nodes, and the Greedy idea is used for node deployment on the real three-dimensional surface model. Its basic idea is as follows:

First of all, the problem to be solved needs to be clarified. The solution goal of this study is to actualize the deployment of sensor nodes under the complex and three-dimensional surface model and maximize the coverage rate while meeting the regional coverage requirements. Therefore, the objective function is:

$$\begin{cases} Coverage \geq P_{pre} \\ Coverage = \frac{sum(count_{p \geq \varphi})}{m \times n} \end{cases} \tag{6}$$

where *Coverage* is the coverage rate under the condition that the preset coverage rate $P_{pre}$ is satisfied, and its calculation formula is the ratio of the number of covered grids to the total number of grids, where the coverage is declared when the probability $p$ is greater than a specified threshold $\varphi$.

Then, the constraint conditions of the algorithm are determined: the deployment position of the candidate nodes cannot exceed the deployment range defined by the area to be deployed. That is, all nodes are deployed on the grid points divided by the surface model. Suppose the sensor coordinate of a node is $(x_i, y_j, z(x_i, y_j))$, and the corresponding constraint condition is:

$$s.t. \begin{cases} x_i \in X & (i = 1, 2 \dots m) \\ y_j \in Y & (j = 1, 2 \dots n) \\ z(x_i, y_j) \in Z \end{cases} \tag{7}$$

Node traversal is carried out according to the idea of the grid-scanning Greedy algorithm.

(1) Calculate the coverage $P(k)$ of any sensor node to its surrounding grid points under the influence of terrain shading and signal attenuation by combining Equations (3)–(5).

(2) Find the node deployment location that can cover the largest number of grid points in the area to be deployed, and add this location to the candidate coverage set *Cov_sel*, as shown in Equation (8):

$$\begin{cases} \arg\max Num(P(k) > \varphi) \\ Cov\_sel = Cov\_sel \cup \{sensor(k)\} \end{cases} \tag{8}$$

(3) Delete the covered grid points from the traversal domain *Trav*. Suppose multiple nodes with the same coverage rate are encountered during the traversal process. In that case, the node with the smallest coordinate is selected for priority deployment until the area to be deployed is fully covered or the predetermined coverage rate is reached, and finally ends the algorithm. The specific formula is shown below:

$$\begin{cases} P(k) > \varphi \\ k \to (p, q) \\ Trav = \sim (Trav(i, j) = 0, Trav(p, q) = 0) \&\& ones(m, n) \end{cases} \tag{9}$$

The implementation of the WSN node deployment algorithm based on real 3D terrain is summarized in Algorithm 1:

---

**Algorithm 1:** WSN deployment based on grid scanning using the Greedy algorithm

---

1 Import of DEM Data.
2 Initialize $R, r, \alpha, \beta, \varphi$.
3 For each surface grid $i$.
4　　Calculate the coverage area corresponding $P(i)$ to the grid point $i$. (3)–(5)
5 End.
6 While ($Coverage < P_{pre}$)
7　　Find the location of the node $k$ to be deployed with the largest coverage. (8)
8　　Add the point $k$ to the candidate coverage set. (8)
9　　Remove the covered grid points from the traversal domain. (9)
10　IF $Num(P(m)) = Num(P(n))$
11　　　IF $m < n$
12　　　　$k = m$;
13　　　Else
14　　　　$k = n$;
15 End.

---

## 5. Simulation Experiments and Analysis

To verify the superiority of the proposed algorithm (GAGS) in solving the 3D complex-surface-coverage problem, we will compare and simulate it with traditional deployment algorithms (PSO, GA, ACO).

First, the coverage of each algorithm is compared through deployment simulations under three surface models of different complexity. Secondly, the coverage attenuation factor, coverage threshold, and preset coverage values in the coverage model were changed and simulated again to test the coverage performance of each algorithm under the coverage model of different parameters. Finally, the iterative time required for different algorithms to complete the deployment plan is compared.

### 5.1. Deployment Simulation under Different Levels of the Complex Surface Model

Deployment simulations on three surface models with different levels of complexity are performed, and the coverage rate that each algorithm can achieve is compared. The deployment parameters of all algorithms are kept consistent, as shown in Table 2.

**Table 2.** Simulation parameters.

| Parameter | Value |
|---|---|
| Projection Area of Monitoring Area | $600 \times 600$ m$^2$ |
| Node Perceived Radius $R$ | 5 m |
| Uncertain perception radius $r$ | 1 m |
| Attenuation factor $\alpha, \beta$ | 0.5 |
| Grid length $g$ | 12 m |
| Coverage threshold value $\varphi$ | 0.8 |
| Preset coverage rate | 1 |

The deployment results are shown in Figures 7–9. The deployment results of the proposed GAGS are more uniform than the other algorithms. Furthermore, to more intuitively demonstrate the superiority of the proposed algorithm in terms of coverage, we compare the coverage rate of different algorithms after deployment, as shown in Figure 10.

Specifically, when deploying nodes using a simple surface model, as shown in Figure 7, GAGS uses 444 nodes to achieve full coverage. When the same number of nodes are deployed, the deployment coverage rate of PSO is 86.12% and GA and ACO were 92.04%, 95.99%, respectively, as shown in Figure 10a.

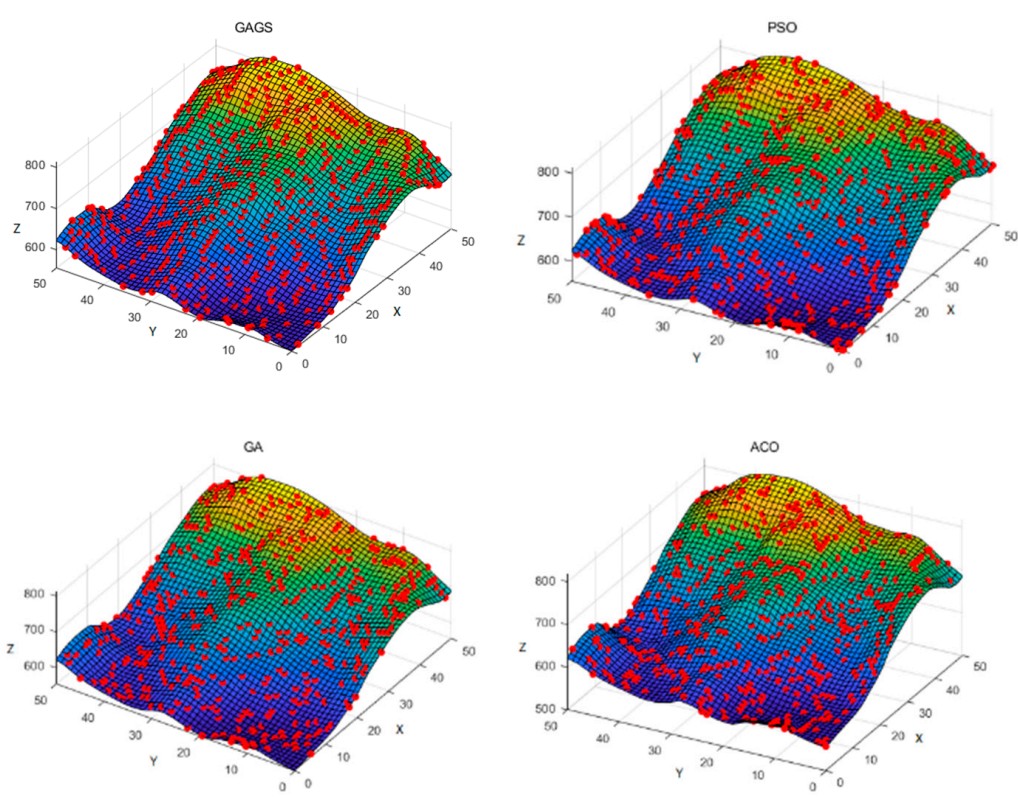

**Figure 7.** Node deployment under simple terrain model.

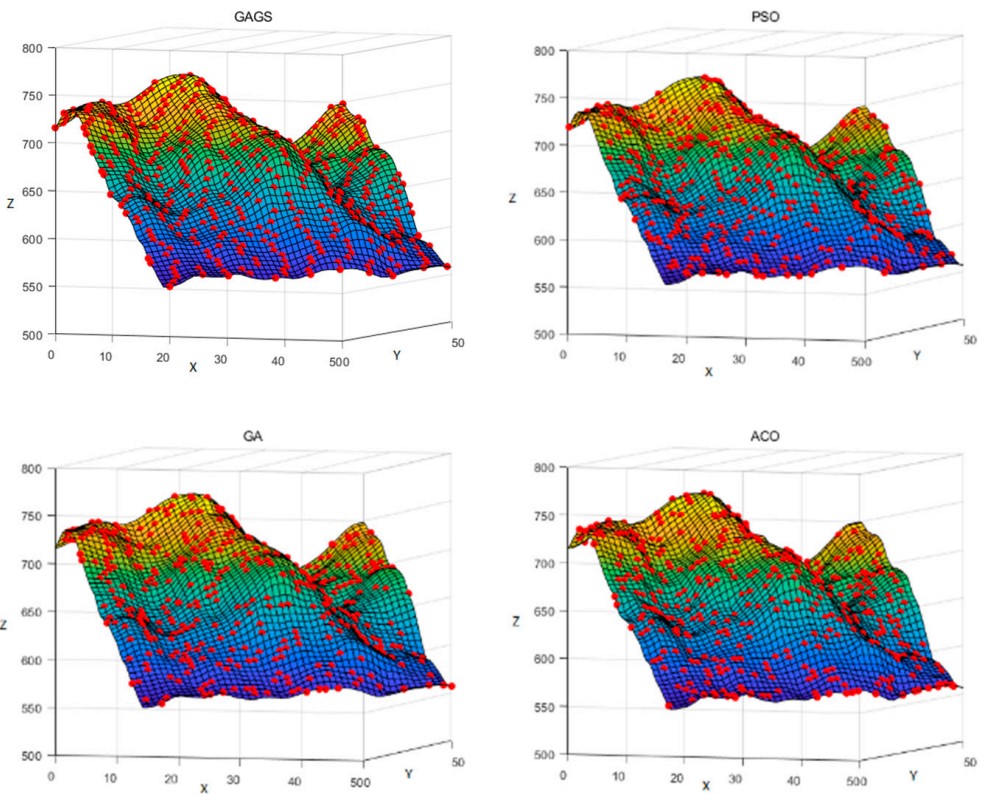

**Figure 8.** Node deployment under the general terrain model.

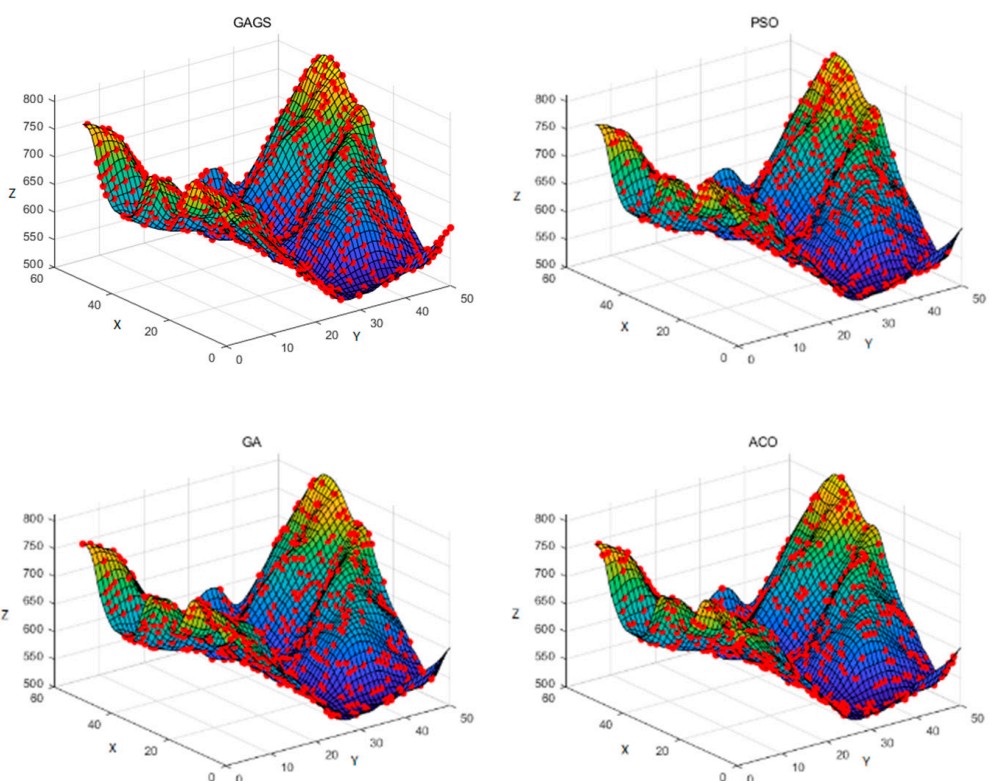

**Figure 9.** Node deployment under complex terrain model.

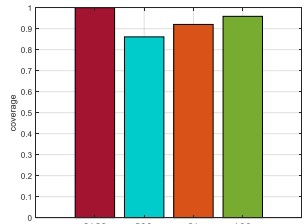

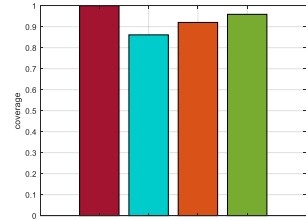

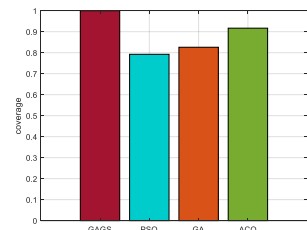

(**a**) Coverage comparison histogram under simple terrain model

(**b**) Coverage comparison histogram under the general terrain model

(**c**) Coverage comparison histogram under complex terrain model

**Figure 10.** Coverage comparison histogram.

Figure 8 presents the deployment simulation of nodes using the general surface model. Since the surface model's projected area is the same, increasing the terrain complexity will increase the surface area and correspond to the number of nodes that need to be deployed. In this case, GAGS uses 486 nodes to achieve full coverage, but when the number of deployed nodes is the same, the deployment coverage of PSO is 83.24%, and GA and ACO are 83.52% and 93.56%, respectively, as is shown in Figure 10b.

Using a complex surface model (Figure 9), GAGS uses 630 nodes to achieve full coverage. In the same number of deployed nodes, the deployment coverage of PSO is 79.33% and GA and ACO are 82.67% and 91.69%, respectively. Compared with the simple terrain model, the coverage descent further, as shown in Figure 10c.

*5.2. Deployment Simulation under Different Coverage Model Parameters*

Take the complex surface model in Figure 9 as an example. The coverage of GAGS under the coverage model with different parameters of the attenuation factor $\alpha$, $\beta$, the threshold $\varphi$, and the value of the preset coverage are analyzed individually.

5.2.1. Deployment Simulation with Different Attenuation Factors

While keeping the other simulation parameters unchanged, the coverage model attenuation factors $\alpha$, $\beta$ were increased from 0.5 to 1. The deployment results of each algorithm are shown in Figure 11.

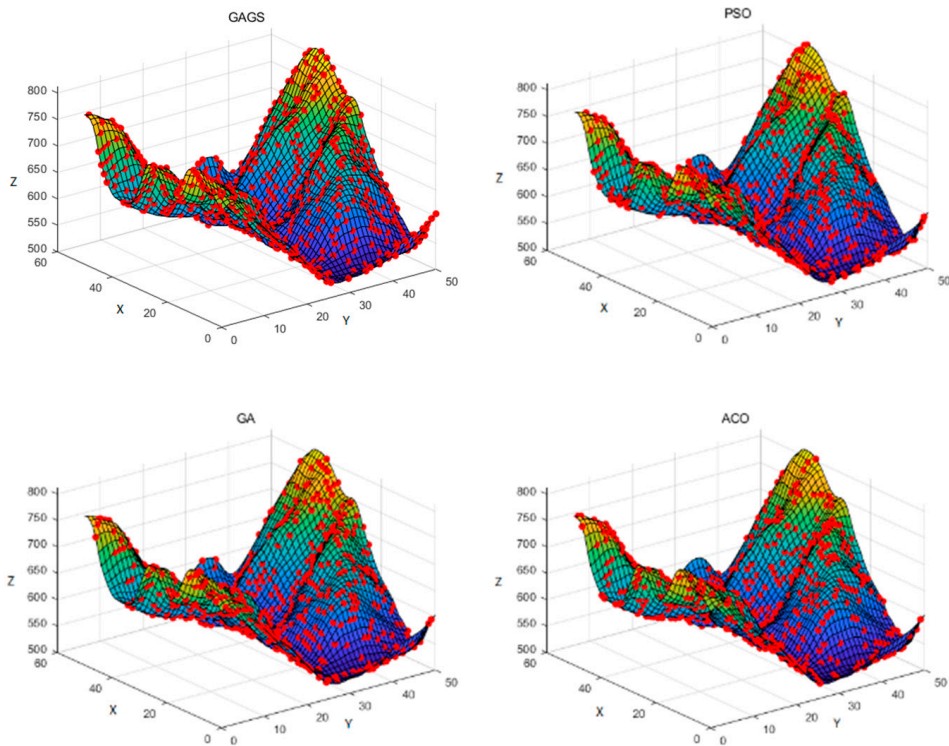

**Figure 11.** Node deployment after increasing the coverage model attenuation factor.

Since the increase of the attenuation factors $\alpha$, $\beta$ will accelerate the attenuation speed of the coverage rate of the perception model, the effective coverage area of each node will be relatively reduced, leading to an increase in the number of WSN nodes to achieve full coverage of the area. The deployment coverage of each algorithm can be seen in Figure 12.

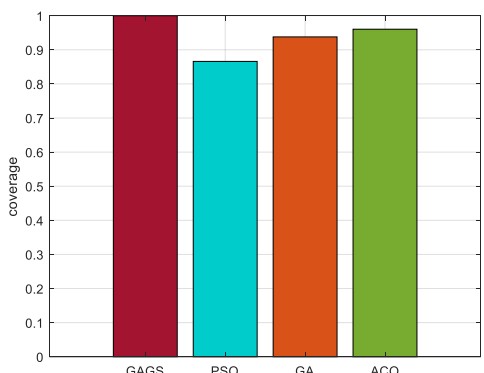

**Figure 12.** Histogram of algorithm coverage comparison after increasing the coverage model attenuation factor.

After increasing the attenuation factors $\alpha$, $\beta$, GAGS achieves full coverage with 684 WSN nodes, while when deploying the same number of nodes, the coverage rate of PSO is only 86.6%, the coverage rate of GA is 93.79%, and that of ACO reached 96.04%. It reveals that the GAGS algorithm still has superiority in coverage performance.

### 5.2.2. Deployment Simulation with Different Attenuation Factors

The coverage threshold reduced $\varphi$ from 0.8 to 0.5 to verify the coverage performance of each algorithm. Other simulation parameters are consistent with Table 2.

The deployment results under different coverage thresholds are presented in Figure 13. It can be seen that reducing the coverage threshold $\varphi$ will expand the coverage of a single node, which means that the number of deployed nodes in the same area decreases. At this time, GAGS achieves full coverage after deploying 375 WSN nodes. The coverage rate of PSO is 86.08%, the coverage rate of GA is 95.44%, and the coverage rate of ACO is 96.11%, under the condition of deploying the same number of nodes. In order to more clearly observe the performance gap of each algorithm, the coverage histogram under the deployment of the same number of nodes is shown in Figure 14.

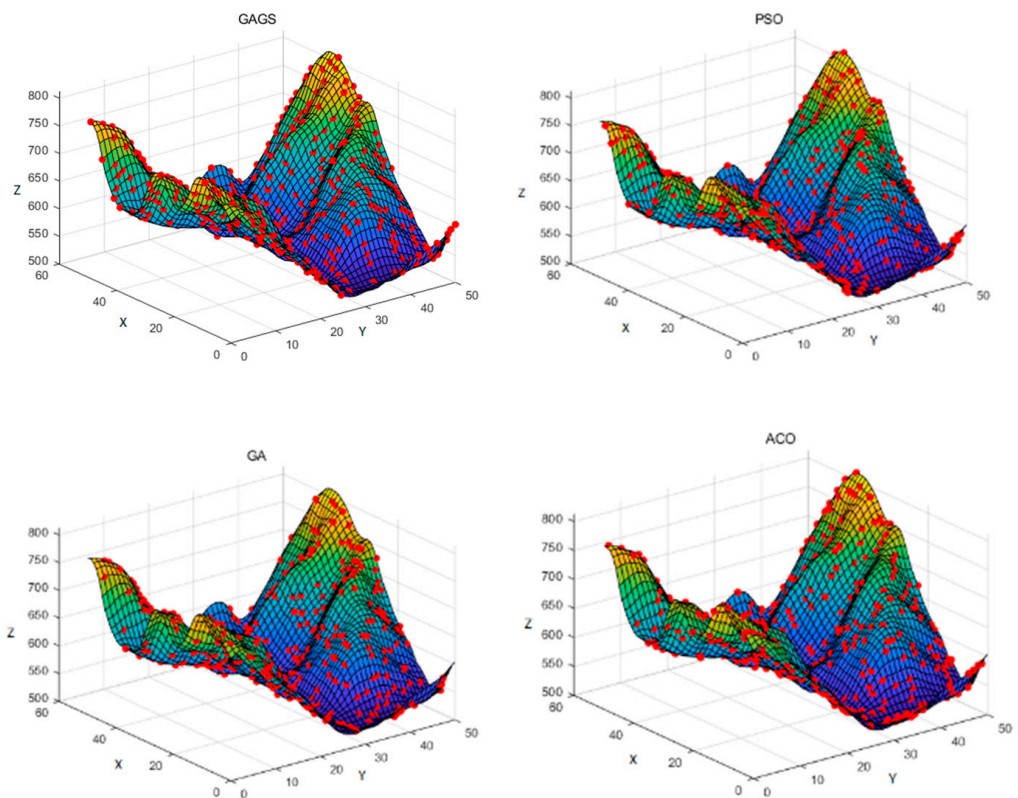

**Figure 13.** Node deployment results after reducing the coverage threshold.

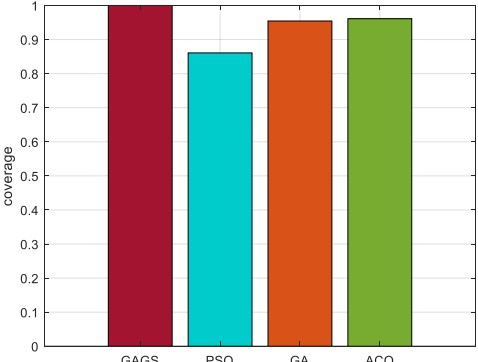

**Figure 14.** Histogram of algorithm coverage comparison after reducing the coverage threshold.

### 5.2.3. Deployment Simulation with Preset Coverage $P_{pre}$

The parameters remained constant, and the only change was the preset coverage rate. The number of nodes that each algorithm needs to deploy under different preset coverage rates are compared in Figure 15.

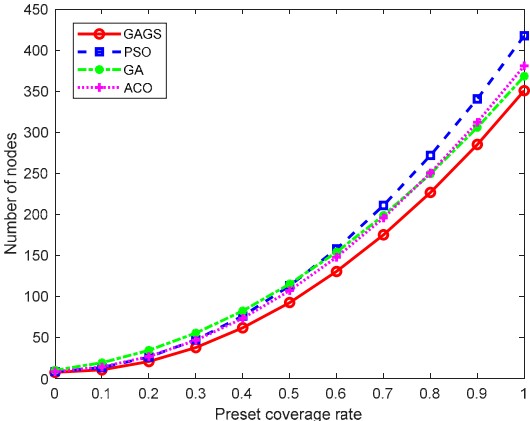

**Figure 15.** Line chart of algorithm coverage ratio comparison under preset coverage ratio.

Obviously, when the preset coverage rate is small, the number of nodes deployed by each algorithm is roughly the same under the premise of meeting the coverage requirements. When the demand for regional coverage is high, the algorithms need to increase the number of nodes to improve the coverage. Through comparison, it is concluded that the GAGS algorithm can meet the coverage requirement under the condition of deploying a lower number of nodes, which is much more cost-effective.

### 5.3. Comparison of Execution Time

The execution time of each algorithm under different surface models was compared. The deployment parameters of all algorithms are shown in Algorithm 1, and the comparison results are shown in Figure 16. With the increase in the complexity of the surface model, the deployment time for all the algorithms has increased. However, the execution time of the GAGS algorithm is significantly shorter than that of the other three algorithms. It shows GAGS can effectively reduce the amount of calculation and the cost.

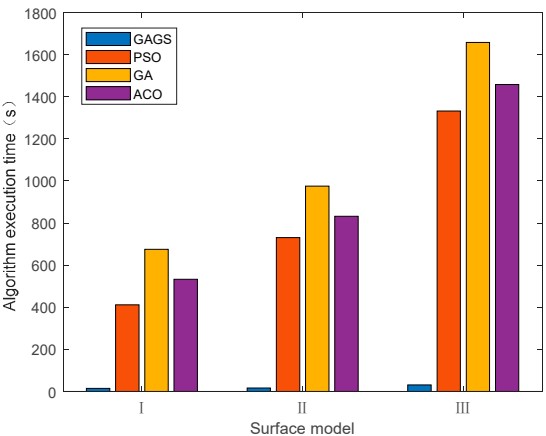

**Figure 16.** Comparison histogram of the execution time required for each algorithm to complete the deployment plan under different surface models.

This is probably because swarm intelligence algorithms (PSO, GA, ACO) are greatly affected by initialization, which is highly random. It needs a large amount of calculation

and has slow convergence in dealing with large-scale combination problems. However, GAGS is an algorithm based on the Greedy idea of grid scanning. It only needs to traverse all grid points once, which significantly reduces the time and space complexity. Moreover, the algorithm deployment scheme is unique and more stable.

## 6. Conclusions

To solve the WSN nodes coverage problem under real 3D terrain, this paper proposes a WSN deployment strategy for real 3D terrain coverage based on the Greedy algorithm with the DEM probability coverage model.

Firstly, DEM data is used to build a three-dimensional real terrain model of the area to be deployed, which can more truly reflect the characteristics of the area environment and make the deployment plan more visible. Secondly, a WSN probability coverage model based on DEM is established, which fully considers the impact of terrain shading and signal attenuation on the coverage model. Finally, a WSN deployment strategy based on the Greedy algorithm of grid scanning is proposed, which effectively improves the coverage of the monitoring area, dramatically reduces the computational complexity, and saves time.

The coverage performance of the proposed GAGS is verified through comparative experiments by comparison with the three traditional deployment algorithms of PSO, GA, and ACO. Experimental results show that for real terrain models with different complexity and coverage models with various parameters, the proposed GAGS can complete the coverage task with fewer nodes and has better coverage performance, faster execution, and more excellent stability.

**Author Contributions:** Conceptualization, W.F. and Y.Y.; methodology, W.F.; software, W.F. and G.H.; validation, W.F., Y.Y.; formal analysis, W.F.; investigation, W.F. and G.H.; resources, W.F. and Y.Y.; data curation, W.F. and G.H.; writing—original draft preparation, W.F.; writing—review and editing, Y.Y. and J.H.; visualization, W.F. and G.H.; supervision, Y.Y.; project administration, Y.Y.; funding acquisition, Y.Y. All authors have read and agreed to the published version of the manuscript.

**Funding:** This research was funded by National Natural Science Foundation of China, grant number 52007156, the Shanxi provincial Key Industry Innovation Chain—Industrial Field Project, grant number 2020ZDLGY15-02.

**Institutional Review Board Statement:** Not applicable.

**Informed Consent Statement:** Not applicable.

**Conflicts of Interest:** The authors declare no conflict of interest.

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
