# Peer review of "WSN Deployment Strategy for Real 3D Terrain Coverage Based on Greedy Algorithm with DEM Probability Coverage Model"

_electronics, doi:10.3390/electronics10162028_

Round 1

Reviewer 1 Report

The paper is well organized and easy to follow. However i suggest some minor modifications: 

-The abbreviations should be well defined (e.g. WSN is  not defined in the introduction)

- the capitalisation should be unified (MNASRI and Du)

-Figure 12-16 should be centred

Reviewer 2 Report

In this paper, an efficient sensor deployment algorithm for a dense Wireless Sensor Network (WSN) is presented and simulated to model in 3D the real surface of a terrain, using the proposed probabilistic coverage model of WSN based on DEM data.

The paper is well written, but following remarks must be made:

  1. Technical issues
  • lines 116-130: Which provider offers the “geospatial data cloud”? From which internet site can be downloaded the DEM files? Which geographical locations are available? Please cite the software used in the text and in the references section;
  • lines 122-123: Change the time of verbs from sentence “Open the DEM file … value data output.”
  • line 162: “… will cause an error between the real coverage and the calculated results.” Can you present a value for that error? If the amplitude order of that error is not big enough, that error can be accepted as it is?
  • 3. Change the yellow color for point 2 into green, brown, etc.
  • lines 174-176: for the DEM model, the grid points are uniformly sampled in space? Please detail how the DEM data points are obtained.
  • line 187: “where is the distance”, which parameter is that distance?
  • to reduce the size of DEM model file during the acquisition process, the difference between coordinates of consecutive grid points can be stored, as for DPCM technique. Actually, this depends on how the DEM data points are obtained.
  • lines 177-179: change the indices for x, y and z: xi with i = 1 to m, yj with j = 1 to n, and zk = zi,j = (xi,yj) with k = 1 to m*n. In equation 7, it is ok.
  • lines 222-223: the text “the text following an equation need not be a new paragraph. Please punctuate equations as regular text.” is from paper template. Please delete it!
  • line 266: (xi, yj, zk)
  • Table 3: Please mention the measuring units for all values.
  • In Figs. 7, 8, 9, 11, 13, the axis for x and y are from 0 to 50, and in Table 3 the projection area is 600*600. Are the same values? Please make the corrections!
  • line 414: “excellent stability” was not defined and analyzed previously. Please detail this performance parameter!

2. Language, Style and Formatting

  • the abbreviations WSN and DEM should be defined in the text, not in the paper’s abstract, when they are firstly used
  • line 59: rephrase the step 3: “The deployment algorithm is employed complete the deployment of nodes”
  • Small caps after punctuation and , instead of . : line 12: “First of all, Actual geographic…”, line 157: “Where P …”, line 287: “coverage problem. We will compare”, line 324: “93.56%, respectively. As is shown”, line 398: “under real 3D terrain. This paper”
  • lines 237-240: the text “the WSN probability attenuation coverage model based on DEM data” is repeated before and after equation 5. Please reformulate it!
  • lines 240-243 must be formatted as Justified
  • line 252: “three-place surface” -> “three-dimensional surface”?
  • line 38, 425: small caps for authors of reference [3]
  • Upper case in Table 3 for “attenuation”, “grid”
  • title of Figure 7 should be on previous page
  • all graphs from Figs 11, 13, should be on the same page
  • rephrase the lines 370-372 for grammar
  • lines 401, 403: “First”, “Second” -> “Firstly”, “Secondly”
  • line 402: “of the rea environment”
